# Effectiveness of a fully immersive virtual reality-based therapeutic exercise programme with altered visual feedback in patients with fibromyalgia: A study protocol for a randomised controlled trial

Carlos Salvador-Huerta[1], Jaime Jordán-López[1], Pedro Azanon-Nogueira[1], Celia García-Lucas[1], Juan J. Amer-Cuenca[2]*, Juan Francisco Lisón[1,3]

1 Department of Biomedical Sciences, School of Health Sciences, Universidad Cardenal Herrera-CEU, CEU Universities, Alfara del Patriarca, Valencia, Spain, 2 Department of Physiotherapy, School of Health Sciences, Universidad Cardenal Herrera-CEU, CEU Universities, Alfara del Patriarca, Valencia, Spain, 3 CIBER of Physiopathology of Obesity and Nutrition (CIBEROBN), Instituto de Salud Carlos III, Madrid, Spain

* juanjoamer@uchceu.es

## Abstract

Fibromyalgia (FM) is a chronic condition characterized by widespread pain, fatigue, and diverse physical and psychological symptoms that significantly impair daily functioning. Latest guidelines advocate for a comprehensive approach to FM management, emphasizing patient education, therapeutic exercise, pharmacological treatments, and psychotherapy. Although exercise remains the primary non-pharmacological strategy supported by robust evidence, its clinical implementation faces several limitations, including poor adherence, fear of movement, pain catastrophizing and proprioceptive deficits. Immersive virtual reality (IVR) has recently emerged as a promising adjunctive tool that not only addresses these limitations but also provides additional therapeutic benefits. Specifically, preliminary studies suggest that incorporating visual feedback manipulation through IVR can effectively modify movement perception, potentially enhancing clinical outcomes. This randomised controlled trial aims to assess the efficacy of a fully immersive virtual reality (FIVR)-based therapeutic exercise programme designed to induce implicit visual illusions, making participants perceive less movement than is performed during structured resistance exercises. Eighty participants diagnosed with FM will be recruited and randomly allocated to either the experimental group, which will engage in a structured resistance exercise regimen combined with FIVR, or the control group, performing identical exercises without FIVR. The primary measure will be the impact of FM on daily life, evaluated using the Revised Fibromyalgia Impact Questionnaire. Secondary outcomes include fatigue, sleep quality, FM symptom severity, health-related quality of life, psychological factors, central sensitization, body perception distortion, lower limb

**Data availability statement:** No datasets were generated or analysed during the current study, as this manuscript reports a study protocol only. Upon trial completion, the de-identified dataset, accompanying data dictionary, and analysis code will be deposited in Zenodo and made publicly available under an open licence; DOIs will be provided upon release.

**Funding:** This work was supported by the University CEU Cardenal Herrera (GIR25/41). The funder had no role in study design, data collection and analysis, decision to publish, or preparation of the manuscript.

**Competing interests:** The authors have declared that no competing interests exist.

strength and handgrip strength, functional mobility, lumbar range of motion, influence of modified visual feedback, behavioural regulation during exercise and overall physical activity levels. This study aims to provide robust evidence regarding the potential benefits of integrating FIVR with therapeutic exercise. Findings could support FIVR as a valuable clinical innovation, potentially yielding superior improvements in daily functioning, physiological outcomes, and psychological well-being compared to traditional exercise interventions alone. ClinicalTrials.gov (NCT06948500). URL: https://clinicaltrials.gov/ct2/show/NCT06948500.

## Introduction

Fibromyalgia (FM) is a condition of unknown etiology, primarily characterized by the presence of chronic (>3 months) and widespread pain affecting various regions of the body [1]. Although widespread pain is the most distinctive clinical feature, FM is a complex, polysymptomatic disorder that includes other core symptoms such as fatigue and sleep disturbances [2]. In addition to these cardinal symptoms, individuals often experience a range of other symptoms and/or dysfunctions, including cognitive impairments [2], regional pain syndromes, autonomic dysfunction [3], psychiatric symptoms and hypersensitivity to external stimuli [1–3].

FM prevalence significantly varies by diagnostic criteria, remaining underrecognized clinically [4]. Despite this, FM is the second most common rheumatic condition after osteoarthritis [1,4], with a global prevalence of 2%–8% [1,5]. FM substantially impairs quality of life and functional capacity, significantly increasing healthcare demand and costs [6]. Annual costs associated with FM have been estimated at €12,993 million in Spain [5]. In the United States, an estimated economic burden of over US$20 billion annually has been reported [7]. Indirect societal costs, primarily from lost productivity, are also considerable; approximately 24.3% of patients stop working within five years post-diagnosis [8].

Given FM's clinical complexity and multisystem involvement, a multidisciplinary approach is essential. The latest European League Against Rheumatism (EULAR) guidelines recommend four main pillars for its management: patient education, therapeutic exercise, pharmacological treatment, and psychotherapy [9]. Non-pharmacological strategies should be prioritized, particularly therapeutic exercise remains the first-line intervention, with strong evidence supporting its benefits [9]. High-frequency, high-intensity transcutaneous electrical nerve stimulation (TENS) has also shown its efficacy in pain reduction being an option for pain relief [10].

Although therapeutic exercise has proven to be an effective intervention in the management of FM, its clinical implementation presents significant limitations. One of the main barriers is poor adherence due to low motivation and high dropout rates, largely attributable to the lack of individualized programmes and the monotony of prescribed routines [11]. In addition, in patients with FM and other disorders characterized by central sensitization, further limiting factors such as kinesiophobia and pain catastrophizing have been identified [12].

Fear of movement, driven by the belief that physical activity will exacerbate pain, leads to avoidance behaviors that contribute to both the maintenance and progression of disability [13]. In this context, kinesiophobia functions not merely as a consequence of pain but as an active perpetuating factor. Maladaptive beliefs about pain, particularly when combined with high levels of kinesiophobia, play a critical role in symptom exacerbation [14]. Recent evidence in individuals with chronic low back pain has shown a similar pattern, with higher pain intensity correlating positively with kinesiophobia, catastrophising and anxiety, and catastrophising emerging as an independent predictor of kinesiophobia [15].

Considering these challenges, innovative therapeutic approaches such as virtual reality (VR) have gained increasing attention. When integrated with therapeutic exercise, VR has shown promise in the management of chronic pain [16]. Recent studies suggest that VR may modulate both pain perception and kinesiophobia through mechanisms such as sensory immersion, cognitive distraction, and multisensory stimulation, positioning it as a valuable tool for pain management [17,18].

VR technologies can be classified into three categories according to their level of immersion: non-immersive, semi-immersive, and fully immersive systems [19]. The degree of immersion directly influences both the user's perceptual and psychological experience, as well as their sense of embodiment, that is, the perceived integration of the self with a virtual avatar within the simulated environment [20].

To date, most VR interventions for FM treatment have employed non-immersive systems [21]. Although offering limited immersion, these platforms demonstrate promising effects in reducing FM's impact on daily life by improving quality of life, symptom burden (fatigue, anxiety, depression), functional capacity and improving adherence [21,22].

In contrast, Immersive virtual reality (IVR) has demonstrated therapeutic advantages over non-immersive systems, primarily due to its capacity to enhance embodiment and deliver real-time multisensory feedback [19,23]. These mechanisms appear to play a pivotal role in clinical outcomes, particularly in pain modulation. Higher levels of immersion are associated with greater attentional distraction and significant reductions in perceived pain [24]. Although limited, existing evidence in FM patients is promising; a recent study integrating IVR with exercise reported significant improvements in pain, kinesiophobia, fatigue, physical activity, and mental quality of life [25]. These findings suggest that increased embodiment and immersion may potentiate the effects of exercise by enhancing cognitive engagement and attenuating central sensitization.

Moreover, visual manipulations in IVR have shown promise in promoting greater movement without increasing perceived effort or pain. Chen et al. [26] demonstrated that patients with chronic neck pain, when exposed to altered visual feedback suggesting they were moving less than they actually were, significantly increased their range of motion (ROM), resulting in clinical improvement. Furthermore, these findings are supported by previous studies, in which the use of fully immersive virtual reality (FIVR) with manipulated visual feedback led to significant reductions in pain, disability, and kinesiophobia in patients with chronic low back pain [27,28].

Therefore, FIVR emerges as an innovative therapeutic approach, not only for modulating pain perception and related FM symptoms but also for improving patients' functionality [25]. Although previous research has extensively investigated VR in non-immersive contexts, mainly through exergames and post-exercise interventions, the therapeutic impact of simultaneously integrating FIVR with exercise remains underexplored. Moreover, the clinical effects of modifying visual feedback during FIVR-based therapeutic exercise programmes for FM patients have yet to be thoroughly examined.

To our knowledge, no previous studies have evaluated the impact of a therapeutic exercise programme based on FIVR that incorporates altered visual feedback on clinical outcomes in patients with FM. In this context, the present protocol describes a research project aimed at assessing the efficacy of an FIVR system that modulates visual input to create the illusion of reduced bodily movement relative to actual motion during exercise training. The experimental group (EG) will engage in an exercise regimen delivered within an immersive virtual environment featuring visual manipulations intended to induce the perception of underperformance. This approach has been shown to promote a greater ROM resulting in clinical improvement. The control group (CG) will perform the same exercise protocol without the integration of FIVR. This

controlled, randomised, parallel clinical trial study will provide evidence regarding the efficacy of FIVR and visual feedback manipulation as an innovative strategy in the management of FM, highlighting their potential to improve the condition's impact on patients' daily lives. Additionally, it will investigate potential improvements in physiological, psychological, and physical outcomes.

## Materials and methods

### Study design

This is a randomised, controlled, two-arm, parallel-group, superiority clinical trial with 1:1 allocation. The protocol follows SPIRIT 2025 and SPIRIT-Outcomes guidance [29]. The participant timeline is shown in Fig 1.

### Ethical approval and registration

This study complies with the Declaration of Helsinki. The protocol was approved by the Research Ethics Committee of CEU Cardenal Herrera University (CEEI25/643). Participation is voluntary, and written informed consent will be obtained before any study procedure. Participants may withdraw from the study at any time without providing a reason and without any penalty or loss of access to current or future healthcare services at the "Asociación Valenciana de Afectados de Fibromialgia" (AVAFI) or any other healthcare facility. Eligible patients will be informed of objectives, procedures, potential risks, and expected benefits before starting the exercise programme. The protocol is registered at ClinicalTrials.gov (NCT06948500). Protocol version v1.0 (18 September 2025. Personal data will be collected by clinical research coordinators and stored in a secure, password-protected database with access limited to authorised personnel.

### Participants: Recruitment and eligibility criteria

This study will enrol 80 participants. Inclusion criteria will consist of adult individuals diagnosed with FM according to any of the classification criteria established by the American College of Rheumatology (ACR), including the 1990, 2010, 2011 or 2016 criteria [30–33]. Additional requirements for meeting the inclusion criteria include the ability to communicate effectively with the research staff, provision of written informed consent demonstrating a clear willingness to participate, and a self-reported pain intensity score of ≥3 on an 11-point Numerical Pain Rating Scale (NPRS-11).

Participants will be excluded if they meet any of the following criteria: (1) presence of comorbidities and/or symptoms that constitute a contraindication for FIVR and exercise-based interventions, including: (i) FIVR-related: history of epilepsy or photosensitive seizures; severe vestibular disorders/vertigo or marked susceptibility to motion sickness/cybersickness; acute symptoms increasing the risk of adverse VR effects at the time of a session (e.g., severe headache/migraine, pronounced dizziness, nausea/vomiting, or disorientation); or severe uncorrected visual impairment preventing safe headset use; and (ii) exercise-related: any unstable or acute cardiopulmonary condition (e.g., unstable angina, uncontrolled arrhythmias, decompensated heart failure, acute myocarditis/pericarditis, acute pulmonary embolism), uncontrolled severe hypertension, or acute systemic illness/fever; (2) presence of medical conditions likely to interfere with outcome assessment or interpretation, such as severe auditory, perceptual, or sensory disorders, or concurrent rheumatological diseases (e.g., rheumatoid arthritis or osteoarthritis); (3) use of medications that are likely to materially interfere with the assessment or interpretation of study outcomes; could potentially affect study results; (4) (i) engagement in another therapeutic physical activity programme during the intervention period; and/or (ii) initiation of new therapies or substantial modifications to existing therapies (including medication or physical therapy) during the study period, which could materially affect study outcomes throughout the study; (5) inability to attend the centre where the interventions will take place.

Participant recruitment strategies will include community outreach initiatives, partnerships with primary care centres, and targeted digital promotion through institutional social media platforms, aimed at ensuring timely and adequate enrolment to achieve the planned sample size.

| Domain/ Measure | Screening | Baseline (t0) | Allocation | Sessions 1-12 | Post (t1) |
|---|---|---|---|---|---|
| **ENROLMENT** | | | | | |
| Eligibility (ACR 1990/2010/2011/2016; VAS ≥3) | X | | | | |
| Informed consent | X | | | | |
| Sociodemographics | | X | | | |
| **ALLOCATION** | | | | | |
| Randomization (stratified by sex and FIQR) | | | X | | |
| **INTERVENTIONS** | | | | | |
| Experimental: Exercise + IVR with altered visual feedback | | | | X | |
| Control: Same exercise without IVR | | | | X | |
| **ASSESMENTS – Primary outcome** | | | | | |
| FIQR total | | X | | | X |
| **ASSESMENTS – Secundary outcomes** | | X | | | X |
| Spanish version of the Multidimensional Fatigue Inventory (MFI-20) | | X | | | X |
| Spanish version of the Pittsburgh Sleep Quality Index (PSQI) | | X | | | X |
| Fibromyalgia Survey Diagnostic Criteria (FSDC) | | X | | | X |
| EQ-5D-5L index and EQ-VAS | | X | | | X |
| Spanish version of the Hospital Anxiety and Depression Scale (HADS) | | X | | | X |
| Spanish version Pain Catastrophizing Scale (PCS) | | X | | | X |
| Spanish version of the 16-item Fear-Avoidance Beliefs Questionnaire (FABQ) | | X | | | X |
| Spanish version of the Tampa Scale for Kinesiophobia (TSK-11) | | X | | | X |
| Central Sensitization Inventory (CSI) | | X | | | X |
| Quantitative sensory testing (QST) | | X | | | X |
| Modified Spanish version of the Fremantle Back Awareness Questionnaire (FreBAQ) | | X | | | X |
| Behavioral Regulation in Exercise Questionnaire- 3 (BREQ-3) | | X | | | X |
| 30-s Sit-to-Stand (STS-30) | | X | | | X |
| Isometric quadriceps strength | | X | | | X |
| Handgrip strength | | X | | | X |
| Timed Up and Go (TUG) | | X | | | X |
| ROM trunk (electro- goniometer) | | X | | | X |
| Spanish short-form version of the International Physical Activity Questionnaire IPAQ-E | | X | | | X |
| **ASSESSMENTS – Session-level monitoring** | | | | | |
| Pain VAS pre / post session | | | | X | |
| RPE (Borg 6–20) and repetitions | | | | X | |
| VR well-being and cybersickness NRS | | | | X | |
| Adverse events | | | | X | |

**Fig 1. SPIRIT schedule of enrolment, interventions, and assessments (t0 = pre-intervention baseline; t1 = immediately after the 6-week programme; Sessions 1–12 = two sessions per week for six weeks).**

Recruitment began on 5 May 2025. As of the time of submission, the study is currently in the recruitment phase. We anticipate the following timeline: A) Participant recruitment will be completed by 10 February 2027; B) Data collection will be completed by December 2027; and C) Results are expected to be published in 2028.

## Randomisation and blinding

Participant recruitment and written informed consent will be obtained by physicians from the "Lifestyle and Health" research group at CEU Cardenal Herrera University. Randomisation will be conducted by an independent researcher using a computer-generated 1:1 sequence with stratified permuted blocks (block size = 2) within strata defined by sex (male/female) and baseline FM severity level (two FIQR-based severity levels derived from clusters 1–2 vs clusters 3–4) [34].

Baseline FIQR will be completed prior to randomisation and scored immediately using item-level responses. FIQR severity level will be assigned at baseline according to Pérez-Aranda et al. using a prespecified scoring tool (e.g., a pre-programmed spreadsheet/REDCap instrument) to enable real-time classification.

Stratification by sex was chosen because FM is substantially more prevalent in women than in men and, given the planned sample size, chance imbalances could otherwise occur. Stratification by FM severity was prioritised to ensure comparability in baseline symptom burden, an important prognostic factor, and to minimise potential shifts in severity distribution arising from recruitment through a single patient association.

Given the target sample size (N = 80) and the expected predominance of women in fibromyalgia, we anticipate that most participants will fall within the female strata, while male strata may be comparatively small. We do not apply a minimum-size rule for strata; if recruitment within a stratum is sparse, the prespecified sequence will still be applied, and final allocation counts by arm overall and within strata will be reported.

Randomisation performance will be monitored descriptively by periodically summarising allocation counts by arm overall and within strata (without examining outcomes), to identify any unexpected allocation irregularities.

Allocation will be concealed with sequentially numbered, opaque, sealed envelopes (SNOSE) prepared off-site with tamper-evident seals and carbonless copies. After baseline assessment, site staff will open the next envelope in numerical order to assign the EG or CG. The randomisation list will be held by an independent data manager with no role in enrolment or assessment. Outcome assessors and the trial statistician will be blinded to group allocation. Participants and intervention providers cannot be blinded due to the nature of FIVR and will be instructed not to disclose allocation during assessments. Assessments will be performed by personnel without access to scheduling or allocation logs; groups will be labelled A/B until database lock. Unblinding will occur only to address serious safety concerns or protocol deviations with safety implications, following written authorisation from the Principal Investigator; all unblinding events will be logged. The randomisation list will remain with the independent data manager.

## Intervention

Participants will complete a six-week therapeutic exercise programme. The only between-group difference is the use of FIVR-based altered visual feedback. All sessions will take place at AVAFI (Valencia, Spain). Frequency: twice weekly, 60 min. Rooms will be climate-controlled, private, and equipped with HTC Vive Pro and exercise implements. All sessions will be delivered by a single licensed, experienced physiotherapist trained in the intervention manual and FIVR safety. Each session comprises:

1. **Warm-up (5 min):** low-intensity mobility and breathing exercises for cervical, lumbopelvic, and peripheral regions, without external load.

2. **Main phase (50 min):** multi-joint exercises for trunk, upper, and lower limbs in standing, seated, supine, and prone positions. Movements synchronised with breathing to enhance motor control. Intensity regulated by the Borg 6–20 scale targeting RPE 13–17. Sessions 1–5: bodyweight only; from session 6: progressive external loads. Per-exercise

volume adjusted to rate of perceived exertion (RPE). At each session we will record NPRS-11, RPE, repetitions, and FIVR-related well-being. These session-level variables will be recorded solely to monitor session delivery and participant response and will not be included in the statistical analysis plan.

3. **Cool-down (5 min):** static stretching of lumbar spine and lower limbs plus diaphragmatic breathing in supine.

The EG follows the same 6-week, twice-weekly, 60-min programme using an FIVR (HTC Vive Pro) that alters visual/proprioceptive input to simulate reduced movement amplitude. The FIVR system applies two visual-proprioceptive illusions: ≈20% arm-shortening during flexion tasks and ≈10% underestimation of extension via a virtual bar, as detailed in S3 Appendix [27,28]. The CG follows the identical protocol without FIVR. Full operational details, session scripts, figures, and videos are provided in S3 Appendix. Intervention manual to enable replication.

## Concomitant care and interventions

Participants will maintain usual care during the 6-week intervention. Permitted: stable pharmacotherapy for FM/comorbidities (≥4 weeks pre-baseline, unchanged until t1); over-the-counter analgesics as rescue per usual care; sleep-hygiene and self-management advice; normal daily physical activity not amounting to a structured programme. Prohibited (from 4 weeks pre-baseline to t1): initiation or dose changes of regular (non-rescue) prescribed analgesics, antidepressants, or anticonvulsants; new physiotherapy, TENS, acupuncture, massage, or other manual therapies; enrolment in structured exercise or VR programmes outside the trial. Participants will avoid rescue analgesics within 24 h before outcome assessments when clinically feasible.

Medication use, including rescue analgesic intake and the timing of the last dose within the preceding 24 h, will be recorded at each assessment timepoint and summarised descriptively by group and timepoint.

## Harms and safety reporting

Adverse events (AEs) will be screened at each session and summarised monthly. AEs will be assessed by the blinded outcome assessor and reviewed by the Principal Investigator monthly and ad hoc. Sessions will be paused or modified for moderate-to-severe cybersickness, presyncope, acute illness, any clinically significant AEs, or at investigator judgement. Participants may be discontinued for safety or major protocol violations. Any serious AEs will be reported to the Research Ethics Committee within 24 hours and will trigger an extraordinary safety review. No data monitoring committee is planned given the minimal-risk intervention; no interim analyses or early stopping rules are planned. All AEs will be systematically monitored, documented, and reported in the final dissemination of findings. Study-related AEs will be managed according to institutional procedures; no additional compensation is planned beyond applicable regulations.

## Outcomes and measurements

Sociodemographic data (including participant sex, age, height, weight, BMI, educational level, physical activity level, employment situation, years since FM diagnosis, smoking status and medication intake) will be collected prior to the implementation of the intervention programme.

**Primary outcome.** The primary outcome, FM-related functional impact, will be assessed using the FIQR [35]. This 21-item instrument evaluates physical function (0–30), overall impact (0–20), and symptom severity (0–50), yielding a total score from 0 to 100, with higher scores indicating greater impairment. The Spanish version demonstrates high internal consistency (α = 0.91–0.95), test–retest reliability (r = 0.82), and strong construct validity [35,36]. Widely considered the gold standard for evaluating FM-related disability [34], the FIQR has also been used to identify clinically meaningful subgroups through cluster analysis, supporting disease stratification and tailored interventions [34]. A change of ≈14% in the total FIQR score has been established as the minimum clinically important difference (MCID) for detecting meaningful improvement [37].

**Secondary outcomes.** Fatigue will be assessed using the Spanish version of the Multidimensional Fatigue Inventory (MFI-20) [38], a validated 20-item self-report instrument measuring five dimensions: general and physical fatigue, reduced activity and motivation, and mental fatigue. Items are scored on a 5-point Likert scale, yielding total scores from 20 to 100, with higher scores indicating greater fatigue. The MFI-20 has shown strong test–retest reliability and validity in Spanish populations, particularly in FM, supporting its clinical and research utility [38].

Sleep quality will be assessed using the validated Spanish version of the Pittsburgh Sleep Quality Index (PSQI), a 19-item self-report measure evaluating sleep over the past month across seven components [39]. Total scores range from 0 to 21, with higher scores indicating poorer sleep. The Spanish PSQI demonstrates strong reliability (α = 0.805; test–retest r = 0.773, p < 0.001). It is widely used in FM research as a reliable measure of sleep dysfunction [40,41].

FM symptoms will be assessed using the 6-item Fibromyalgia Survey Diagnostic Criteria (FSDC), aligned with the 2016 ACR revision [33]. It includes the Widespread Pain Index (WPI; 0–19) and the Symptom Severity Scale (SSS; 0–12), evaluating pain distribution and core symptoms (fatigue, nonrestorative sleep, cognitive issues), plus additional somatic symptoms. The total score (0–31) reflects FM symptom burden, with higher scores indicating greater severity. The FSDC has been used in RCTs with Spanish populations, supporting its validity in this context [42].

Health-related quality of life will be assessed using the Spanish EQ-5D-5L, a validated self-reported tool measuring five dimensions on five severity levels [43]. Responses yield an index score (−0.5 to 1.0), with lower values indicating worse health, calculated using the Spanish value set [43]. In addition, patients will rate their perceived health-related quality of life using the EQ-5D-5L visual analogue scale (EQ VAS, 0–100), where 0 represents the worst and 100 the best imaginable health state. Widely used in FM research [44], a change of 0.03–0.07 in the index is generally considered clinically meaningful [45].

Anxiety and depression will be assessed using the Spanish version of the Hospital Anxiety and Depression Scale (HADS) [46], a 14-item self-report tool comprising two subscales, HADS-Anxiety and HADS-Depression, each scored 0–21. Items are rated on a 4-point Likert scale (0–3), with higher scores indicating greater severity. Scores of 0–7 are normal, 8–10 suggest possible cases, and ≥11 indicate clinically significant symptoms [47]. The scale has shown high internal consistency and strong validity in both general and FM populations [48].

Pain catastrophizing will be assessed using the Spanish version of the 13-item Pain Catastrophizing Scale (PCS) [49,50], which evaluates the frequency of catastrophic thoughts and feelings related to pain across three subscales: rumination, magnification, and helplessness. Items are rated from 0 to 4, yielding a total score of 0–52, where higher scores indicate greater catastrophizing. The PCS has shown good psychometric properties and sensitivity to change [50]. A total score of >30 will be considered clinically significant [51].

Fear-avoidance beliefs will be assessed using the Spanish version of the 16-item Fear-Avoidance Beliefs Questionnaire (FABQ) [52,53], scored on a 7-point Likert scale (0–6), where higher scores indicate stronger avoidance beliefs. FABQ comprises two subscales: Work (FABQ-W; range 0–42) and Physical Activity (FABQ-PA; range 0–24), evaluating the perceived influence of occupational and physical activity on pain. The Spanish version has demonstrated excellent psychometric properties [53].The MCID for the FABQ-PA subscale is estimated at 6 points [54].

Kinesiophobia will be assessed using the Spanish version of the Tampa Scale for Kinesiophobia (TSK-11), an 11-item self-report measure of fear of movement and (re)injury [55]. Items are scored on a 4-point Likert scale (1–4), with total scores ranging from 11 to 44; higher scores indicate greater kinesiophobia. The TSK-11 has demonstrated strong psychometric properties and validity in Spanish chronic pain populations, including FM [55,56].

Central sensitization will be assessed using the Central Sensitization Inventory (CSI), a validated 25-item self-report measure [57]. Scores range from 0 to 100, with values ≥40 indicating the presence of a central sensitization syndrome (CSS), including FM [58]. Severity levels include subclinical (0–29), mild (30–39), moderate (40–49), severe (50–59), and extreme (60–100) [59]. The Spanish version demonstrates excellent test–retest reliability (ICC = 0.82–0.91) and is widely used in chronic pain research [60,61].

 

Quantitative sensory testing (QST) will be used to characterise central nociceptive processing through pressure pain threshold (PPT), temporal summation (TS) and conditioned pain modulation (CPM). Baseline PPTs will be assessed by algometry at two sites: the dorsal aspect of the distal phalanx of the index finger and a point 5 cm to the right of the L3 spinous process. PPT will be defined as the lowest pressure that elicits the first sensation of pain under standardised conditions and will be measured with an analogue Wagner algometer (Wagner Instruments, Greenwich, CT, USA) with a 1 cm² probe, applying pressure at 1 kg/cm²/s; three consecutive PPT values will be obtained at the index finger at 30-second intervals and their mean will be used as the test stimulus in the TS and CPM protocols [62,63]. After a 2-minute rest, TS will be evaluated by delivering 10 pressure pulses at the individual index-finger PPT (≈2 kg/s), while participants rate the pain intensity of the first and tenth pulses on the NPRS-11; TS will be calculated as the difference between these ratings, with positive values indicating increased pain facilitation [15,63,64]. Following a further 5-minute interval, CPM will be assessed by repeating the TS procedure in the presence of a conditioning stimulus induced by an occlusion cuff on the left arm, inflated at 20 mmHg/s until the first sensation of pain is reported, maintained for 30 seconds and then adjusted to a pain intensity of 3/10 on the NPRS; CPM will be defined as the difference between PPT during conditioning and base-line PPT at the index finger (CPM = PPT_post – PPT_pre), with positive values reflecting effective endogenous inhibition and lower or negative values indicating reduced inhibitory capacity [63,65].

Body perception distortion will be assessed using the Spanish FreBAQ-S [66], a validated 9-item scale (0–36) originally designed to evaluate altered body image and tactile awareness in chronic musculoskeletal pain [67]. A slightly adapted version replacing "back" with "body" will be used, as in prior FM studies [68]. FreBAQ-S shows solid psychometrics: unidimensional structure (CFI = 0.97; TLI = 0.96; RMSEA = 0.06), high internal consistency (α = 0.82), test–retest reliability (ICC = 0.78), and convergent validity with pain, disability, kinesiophobia, catastrophizing, and CS [66].

Lower limb strength and endurance will be assessed using the 30-Second Sit-to-Stand Test (STS-30), a functional test that evaluates an individual's ability to repeatedly rise from a chair without using their arms over a 30-second period. The total number of repetitions performed within the time frame serves as an indicator of functional capacity, with higher scores reflecting better performance. The test has demonstrated high test–retest reliability (ICC = 0.84–0.96), and it has been validated in studies involving populations with FM and older adults [69–71].

Functional mobility will be assessed using the Timed Up and Go (TUG) Test, a widely used clinical tool for evaluating gait speed, dynamic balance, and overall functional capacity [72]. The test involves timing how long it takes the participant to rise from a chair, walk three metres, turn 180 degrees, return, and sit down again. It has demonstrated high test–retest reliability, with intraclass correlation coefficients (ICCs) of 0.935 using a manual stopwatch and 0.955 with an automated timer.

Isometric quadriceps strength will be assessed using handheld dynamometry, a widely applied technique for quantify-ing neuromuscular function in individuals with FM [71]. A portable dynamometer will be positioned at the level of the ankle with the knee flexed at 90°, to measure the participant's maximal voluntary isometric contraction (MVIC) of the quadriceps muscle [73]. Assessing quadriceps strength serves as a key indicator of lower-limb functionality and the risk of disability in this population. The test–retest reliability has been reported to range between 0.85 and 0.96 (ICC), and its use has been validated in previous studies, providing an objective measure of muscular response to exercise-based therapeutic inter-ventions [73].

Handgrip strength will be measured with a Jamar hydraulic dynamometer following Hamilton et al. protocol [74]. Par-ticipants will sit with feet flat, shoulder adducted/neutrally rotated, elbow at 90°, and wrist/forearm neutral. Three maximal trials with verbal encouragement and ≥4-minute rests will be averaged. This reliable method is widely used in FM [70,71].

Lumbar flexion and extension ROM and the influence of modified visual feedback will be assessed under three exper-imental conditions. Each participant will perform three repetitions of each movement in: (i) a control condition without VR, (ii) an understated condition (E−), in which VR feedback portrays the participant's movement as reduced relative to actual performance, and (iii) an overstated condition (E+), in which VR feedback presents movement as amplified compared

with reality. Lumbar motion, including the flexion-evoked pain threshold (defined as the angle at which the participant first reports pain during forward bending), will be recorded in degrees using a 3-Space Fastrack electrogoniometer connected to a Windows 10-based computer. For each condition, the mean ROM across the three repetitions (absolute data) will be calculated. To account for inter-individual variability, ROM values in the E+ and E− conditions will also be expressed as a percentage of the control condition (relative data). The 3-Space Fastrack system is a validated electro-goniometer for assessing lumbar mobility in patients with low back pain; it uses two motion sensors placed over the T12 and S1 spinous processes and has demonstrated high reliability [75,76].

Behavioural regulation in exercise will be assessed with the Behavioural Regulation in Exercise Questionnaire-3 (BREQ-3), a 24-item self-report tool based on Self-Determination Theory [77]. It measures six motivation types and overall self-determination. The validated Spanish version shows good reliability (α = 0.81 for intrinsic, 0.70 for amotivation) and strong factorial validity [78]. In FM, motivation is key for exercise adherence, impacting function and quality of life, making BREQ-3 essential for evaluating motivational profiles in therapeutic contexts.

Participants' experience with FIVR will be assessed using NPRS-11for both cybersickness intensity and satisfaction, in order to monitor the tolerability and acceptance of the intervention. FIVR has shown promise as an effective tool for managing chronic pain, including FM, and its application in the present study will be closely monitored to ensure participant comfort. Additionally, NPRS-11 will be used to assess pain intensity immediately before and after each session, providing an objective measure of the intervention's short-term impact on perceived pain.

Physical activity will be measured using the IPAQ-E, the Spanish short-form version of the International Physical Activity Questionnaire [79]. This validated tool includes seven items assessing sedentary behaviour, walking, and moderate-to-vigorous activities over the past week. Total physical activity is expressed in MET-min/week, categorizing individuals into low, moderate, or high activity levels [79].

All AEs that may occur during the study will be systematically monitored, documented, and reported in the final dissemination of findings. During each session, the number of repetitions in every exercise set, perceived exertion (Borg scale), and pain level (pre- and post-session) will be recorded. The study design and progression of patients through the study protocol are outlined in Fig 2.

## Statistical methods

Sample size was calculated using G*Power 3.1.9.2 based on standard methodological guidelines. The primary outcome is the impact of FM on patients' lives, assessed with FIQR. The minimum clinically important difference between groups was set at ≈14% based on previous research [37], which corresponds to an absolute between-group difference of 9.1 FIQR points assuming a baseline mean of 65. With α = 0.05 (two-sided), 80% power, and an assumed SD of 20, the trial was powered to detect a baseline-adjusted post-intervention between-group difference using an ANCOVA framework. GPower inputs were: test family = F tests; statistical test = ANCOVA: fixed effects, main effects and interactions; number of groups = 2; number of covariates = 1 (baseline FIQR); effect size f = 0.23 (equivalent to d ≈ 0.46). This yielded 32 participants per group (64 total), which was increased to 80 participants to account for a potential 20% dropout rate. This calculation aligns with the post-intervention between-group contrast estimated from the prespecified two-time point LMM.

Analyses will follow an intention-to-treat (ITT) approach, including all randomised participants in the groups to which they were allocated. Primary and secondary outcomes will be analysed using linear mixed-effects models (LMMs), including fixed effects for group, time, and the group × time interaction, with a random intercept for participants to account for within-subject correlation. Outcomes are assessed at baseline and post-intervention only. Models will be adjusted for prespecified covariates (sex and FM severity level), with covariate adjustment intentionally restricted to this small prespecified set to minimise overfitting and preserve model stability given the planned sample size. Baseline outcome values are accommodated within the two-time point longitudinal model, rather than entered as separate baseline covariates. Other

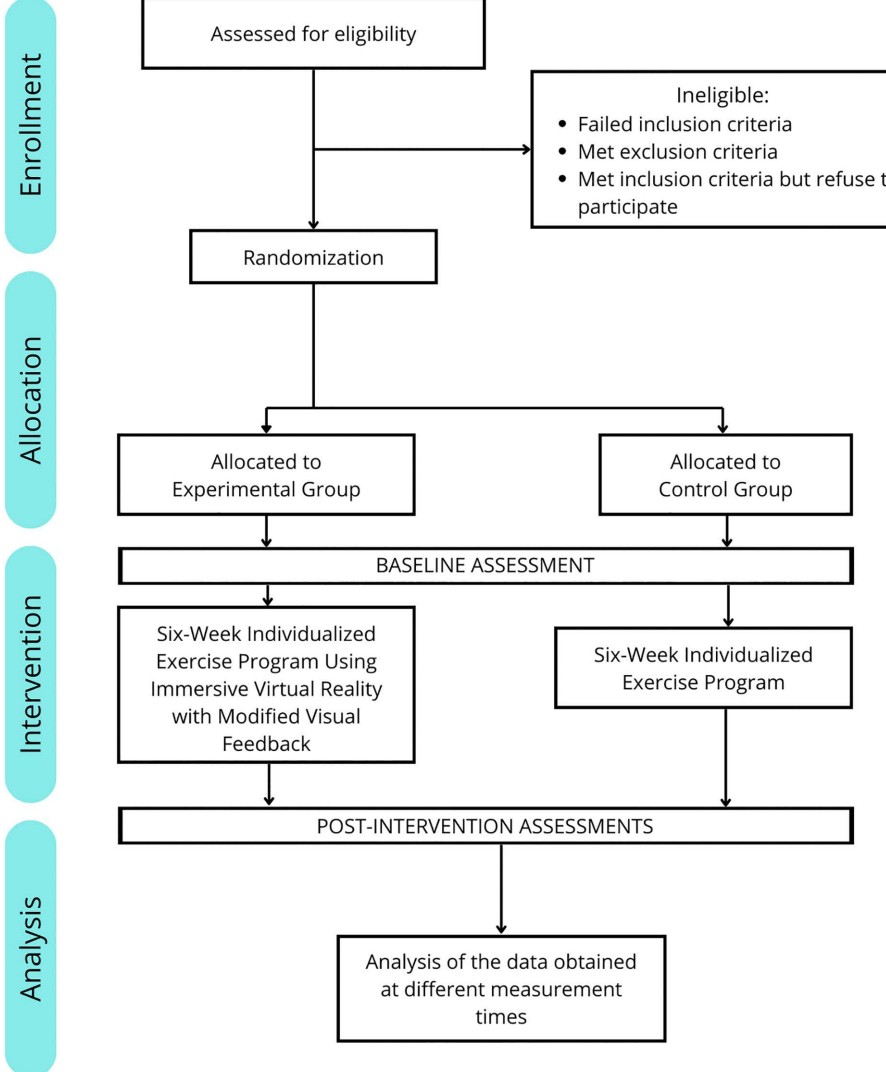

**Fig 2. Protocol timeline implemented for the study.**

baseline variables (e.g., medication use and other clinical characteristics) will be summarised descriptively and explored only in prespecified sensitivity analyses if substantial baseline imbalance is observed; covariates will not be selected data-driven based on statistical significance.

The primary estimand will be the between-group treatment effect from baseline to post-intervention for FIQR. For the two-time point setting, we will use a constrained longitudinal data analysis (cLDA) parametrisation, in which baseline means are constrained to be equal and the treatment effect (difference in change) is expressed as the post-intervention between-group contrast; this aligns with the baseline-adjusted post-intervention treatment effect used for the sample size calculation. In simpler terms, the cLDA approach assumes that both groups start from the same baseline, which is guaranteed by the randomization process, and focuses on comparing the groups at the end of the 6-week program. This allows for a more intuitive assessment of whether the integrated FIVR (EG) provides a specific therapeutic benefit over the same exercises performed without the virtual reality headsets (CG).

Given the large number of secondary outcomes, all secondary outcomes will be treated as exploratory and will be reported with effect estimates and 95% confidence intervals, without confirmatory inferential claims. When feasible, standardised effect sizes (Hedges' g) or partial eta-squared ($\eta p^2$) will be reported. Medication use and rescue analgesic intake will be recorded at each assessment and explored in sensitivity analyses if substantial baseline imbalances are observed.

Missing data will be handled primarily via maximum likelihood estimation within the LMM framework under the missing-at-random (MAR) assumption, utilizing all available observations. To support this assumption, we will document reasons for missingness and compare baseline characteristics between participants with complete and incomplete follow-up. The MAR assumption is considered plausible in this clinical context because dropout in fibromyalgia exercise trials is often related to baseline factors already accounted for in our model, such as initial pain intensity or functional impairment. However, a limitation of this assumption is the possibility of Missing Not At Random (MNAR) data, where a participant might withdraw due to factors not captured by baseline variables, such as a lack of perceived improvement or a specific individual intolerance to the FIVR technology. As a sensitivity analysis, we will perform multiple imputations (FCS/MICE, m = 30), incorporating outcomes at both time points, baseline values, prespecified covariates (sex and FM severity), time indicators, and relevant auxiliary variables predictive of missingness or outcomes.

Quantitative sensory testing (QST) outcomes, including PPT, TS, and CPM, will be analysed as continuous variables using the prespecified LMM. QST will be obtained unilaterally on the dominant side at the hand and lumbar region. PPT will be summarised per site as the mean of three trials. TS and CPM will be derived from pain ratings collected at the first and last repetition (computed as last minus first). If a participant declines testing or a measure cannot be completed, the outcome will be treated as missing. If the maximum limit of the algometer is reached, the value will be recorded at the device maximum and flagged to assess the impact of these ceiling values. Model diagnostics will be inspected, and appropriate transformations (e.g., log-transformation for PPT) or sensitivity analyses excluding extreme values (>3 SD from the mean) will be conducted if assumptions are not met.

In addition to the ITT analyses, a prespecified per-protocol (PP) analysis will be reported as a sensitivity analysis. The PP set will include participants who attend at least 75% of the scheduled sessions (9 of 12 sessions over 6 weeks) without major protocol deviations, using the same LMM approach as the primary analysis. Baseline characteristics will be summarised descriptively by group, and intervention effects will be reported with 95% confidence intervals. Statistical significance for the primary outcome (FIQR) will be assessed at $p < 0.05$ (two-sided). Adverse events will be summarised by group (number of participants with ≥1 event, total events, severity, and relatedness). Between-group differences in incidence proportions will be estimated using risk ratios and risk differences with 95% confidence intervals; where appropriate, Fisher's exact test will be used to obtain p-values given expected small counts.

## Data management

Participant data will be managed using anonymised identification codes, with all entries stored in a secure, password-protected database. Data access will be limited exclusively to an independent data manager with no conflicts of interest or involvement in other aspects of the study. Given the low-risk nature of the research, the establishment of a Data Monitoring Committee has been deemed unnecessary. Any relevant amendments to the study protocol occurring throughout the course of the trial will be duly reported to both the trial registry and the journal where the protocol is published.

## Patient and public involvement

Patients or the public were not involved in the design, conduct, or reporting of this trial. After completion, participants will receive a plain-language results summary, and a public summary will be posted on ClinicalTrials.gov and shared via AVAFI channels.

## Discussion

To the best of our knowledge, this will be the first randomised clinical trial to evaluate the efficacy of a FIVR intervention incorporating real-time feedback modulation during a therapeutic exercise programme for individuals with FM. This protocol addresses a relevant clinical gap in FM management, where exercise is a first-line, evidence-based intervention yet implementation is hindered by poor adherence and limited effectiveness for pain and fatigue. Patients often struggle to engage in conventional programmes due to fear of symptom exacerbation, low motivation, and poor tolerance, underscoring the need for approaches that enhance acceptability and long-term adherence.

VR integration into exercise prescription has shown promise [80,81]. The FM literature has predominantly used non-immersive VR [21], with encouraging results [21,22,82]. Although FIVR in FM remains largely unexplored, evidence indicates FIVR outperforms non-immersive VR for chronic pain [83–85]. FIVR can provide analgesia, increase engagement, and enable customisable experiences; in FM it has improved pain, kinesiophobia, fatigue, physical activity, and HRQoL-mental vs controls [25].

However, prior work has key methodological limits. Gulsen et al. (2020) had a small sample (n = 20), delivered FIVR after exercise, and limited exposure to 20 min/session, restricting generalisability and dose–response inference and possibly influencing outcomes.

This protocol tests a 60-min FIVR intervention with manipulated visual feedback delivered concurrently with a structured TEP. The system dynamically underrepresents actual movement amplitude, encouraging movement beyond perceived limits and promoting greater ROM and functional gains. This perceptual manipulation seeks to enhance motor learning and symptom modulation without increasing perceived exertion or pain, while sustained in-session engagement may support adherence and clinical effectiveness.

## Supporting information

**S1 Checklist. SPIRIT 2025 checklist.**
(DOCX)

**S1 Protocol. Spanish protocol.**
(DOCX)

**S2 Protocol. English protocol.**
(DOCX)

**S3 Appendix. Intervention.**
(DOCX)

**S4 Document. Ethics approval (EN).**
(PDF)

**S5 Document. Ethics approval (ES).**
(PDF)

**S6 Checklist. Human participants research checklist.**
(DOCX)

## Acknowledgments

We thank all participants and the Asociación Valenciana de Afectados de Fibromialgia (AVAFI) for recruitment support and access to facilities. We also thank the staff of CEU Cardenal Herrera University for administrative support.

## Author contributions

**Conceptualization:** Carlos Salvador-Huerta, Jaime Jordán-López, Juan Francisco Lisón.

**Data curation:** Carlos Salvador-Huerta.

**Formal analysis:** Carlos Salvador-Huerta, Jaime Jordán-López, Juan J. Amer-Cuenca, Juan Francisco Lisón.

**Investigation:** Carlos Salvador-Huerta, Jaime Jordán-López, Pedro Azanon-Nogueira, Celia García-Lucas, Juan J. Amer-Cuenca, Juan Francisco Lisón.

**Methodology:** Carlos Salvador-Huerta, Jaime Jordán-López, Juan J. Amer-Cuenca, Juan Francisco Lisón.

**Project administration:** Carlos Salvador-Huerta, Jaime Jordán-López, Juan J. Amer-Cuenca, Juan Francisco Lisón.

**Resources:** Juan J. Amer-Cuenca, Juan Francisco Lisón.

**Supervision:** Juan J. Amer-Cuenca, Juan Francisco Lisón.

**Validation:** Juan J. Amer-Cuenca, Juan Francisco Lisón.

**Visualization:** Carlos Salvador-Huerta.

**Writing – original draft:** Carlos Salvador-Huerta.

**Writing – review & editing:** Carlos Salvador-Huerta, Jaime Jordán-López, Pedro Azanon-Nogueira, Celia García-Lucas, Juan J. Amer-Cuenca, Juan Francisco Lisón.

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
