## [Decision Letter · Decision Letter 0]

24 Feb 2026

PONE-D-25-62476Effectiveness of an immersive virtual reality-based therapeutic exercise programme with altered visual feedback in patients with fibromyalgia: A study protocol for a randomised controlled trialPLOS One

Dear Dr. Amer Cuenca,

Thank you for submitting your manuscript to PLOS ONE. After careful consideration, we feel that it has merit but does not fully meet PLOS ONE’s publication criteria as it currently stands. Therefore, we invite you to submit a revised version of the manuscript that addresses the points raised during the review process.

We look forward to receiving your revised manuscript.

Kind regards,

Hui-Juan Cao, Ph.D.

Academic Editor

PLOS One

Journal Requirements:

6. Thank you for stating the following financial disclosure:

This work was supported by grants from the University CEU Cardenal Herrera (GIR25/41).

7. Please amend your authorship list in your manuscript file to include author Juan Francisco Lisón Párraga

8. Please amend the manuscript submission data (via Edit Submission) to include author Juan Francisco Lisón

9. Please amend either the abstract on the online submission form (via Edit Submission) or the abstract in the manuscript so that they are identical.

10. We note that there is identifying data in the Supporting Information file <S4 Document. Ethics approval (EN).pdf, S5 Document. Ethic approval (ES).pdf>. Due to the inclusion of these potentially identifying data, we have removed this file from your file inventory. Prior to sharing human research participant data, authors should consult with an ethics committee to ensure data are shared in accordance with participant consent and all applicable local laws.

-Location data

Reviewers' comments:

Reviewer's Responses to Questions

**Comments to the Author**

1. Does the manuscript provide a valid rationale for the proposed study, with clearly identified and justified research questions?

Reviewer #1: Yes

Reviewer #2: Yes

2. Is the protocol technically sound and planned in a manner that will lead to a meaningful outcome and allow testing the stated hypotheses?

Reviewer #1: Yes

Reviewer #2: Yes

3. Is the methodology feasible and described in sufficient detail to allow the work to be replicable?

Reviewer #1: Yes

Reviewer #2: Yes

4. Have the authors described where all data underlying the findings will be made available when the study is complete?

Reviewer #1: Yes

Reviewer #2: Yes

5. Is the manuscript presented in an intelligible fashion and written in standard English?

Reviewer #1: Yes

Reviewer #2: Yes

6. Review Comments to the Author

You may also provide optional suggestions and comments to authors that they might find helpful in planning their study.

Reviewer #1: Overall Assessment

This Study Protocol “Effectiveness of an immersive virtual reality-based therapeutic exercise programme with altered visual feedback in patients with fibromyalgia: A study protocol for a randomised controlled trial” presents a well-designed, two-arm, parallel-group RCT evaluating the effectiveness of an immersive virtual reality (IVR) therapeutic exercise programme incorporating altered visual feedback for individuals with fibromyalgia (FM). From a statistical and methodological perspective, the planned trial is conceptually sound, adheres to SPIRIT 2025 guidance, and incorporates several strengths: prespecified outcomes, a justified sample-size calculation, a detailed randomisation and blinding strategy, and appropriate use of covariate-adjusted models.

Nevertheless, several aspects would benefit from clarification or refinement to ensure full transparency, reproducibility, and appropriateness of statistical analyses, especially considering the complexity of the intervention and the breadth of outcomes being collected.

My major and minor statistical comments follow.

Major Comments

1. Sample Size Justification: Effect Size Assumptions Need Clarification

The sample size uses an assumed SD = 20 and an MCID of ≈14% on the FIQR to detect between-group differences using ANCOVA. However:

• The protocol does not state the absolute expected mean FIQR score on which the 14% MCID is computed.

• It is unclear whether the authors powered the study for between-group mean difference or group × time interaction.

Recommendation:

Explicitly report the assumed baseline FIQR mean, the corresponding absolute MCID value, the expected post-intervention difference, and whether the model used in the power calculation aligns with the final analysis approach (ANCOVA vs. repeated-measures ANCOVA). Provide the full numeric parameters used in G*Power for reproducibility.

2. Choice of Analysis Method: Repeated-Measures ANCOVA May Not Be Appropriate

The plan states that two-way repeated-measures ANCOVA will be used, adjusting for covariates (sex, FIQR cluster, baseline scores).

Two key concerns:

• Repeated-measures ANCOVA is not a standard or well-defined model.

• Adjusting for baseline within a repeated-measures framework typically leads to statistical redundancy and may violate model assumptions.

Recommendation:

Use a linear mixed-effects model (LMM), which is the standard for RCTs with repeated measures, missing data handled under MAR, and covariate adjustment. LMMs also allow random intercepts (and slopes if appropriate), avoid sphericity assumptions, and appropriately model within-subject correlation.

3. Multiple Secondary Outcomes: No Adjustment Strategy Specified

The study includes an exceptionally large set of secondary outcomes (psychological scales, sensory testing, mobility tests, physiological measures, IVR tolerance, motivation, and more).

Given the number of endpoints, there is a substantial risk of false positives.

Recommendation:

Specify a plan for multiplicity control, such as:

• Designating key secondary outcomes a priori and controlling FDR across them, or

• Applying hierarchical testing, or

• Clearly stating that secondary outcomes are exploratory without inferential claim.

PLOS ONE’s policies permit exploratory outcomes, but this must be stated unambiguously.

4. Handling of Missing Data: Multiple Imputation Needs Additional Specification

The protocol states that multiple imputations will be used assuming data missing at random (MAR).

Issues requiring clarification:

• The number of imputations is not specified.

• The imputation model variables are not described.

• MAR assumption justification is absent.

• For longitudinal outcomes, multilevel imputation or model-based approaches are usually preferred.

Recommendation:

Specify:

1. Number of imputations (≥20 recommended).

2. Variables included in the imputation model (all predictors, outcomes, and auxiliary variables).

3. Whether imputation will be performed separately by randomized group.

4. Whether imputation will incorporate repeated-measures structure (e.g., MICE with time indicators).

5. Randomisation: Stratification and Block Sizes Are Appropriate but Require Clarification

Randomisation uses stratified block permutation across eight strata (sex × FIQR severity clusters) with block sizes 2–4.

Concerns:

• FIQR cluster derives from Pérez-Aranda et al., but the method for real-time assignment of clusters in the trial is not fully described.

• The number of strata (8) for a target N=80 may result in unstable allocation within small strata.

Recommendation:

Report:

• The decision process and data required to assign FIQR cluster at baseline.

• Expected sample per stratum and any minimum-size rule.

• Whether randomisation performance (e.g., imbalance assessment) will be monitored.

6. Quantitative Sensory Testing (QST): Statistical Plan Requires Additional Detail

The QST section is detailed in measurement procedures, but statistical treatment lacks specification (PPT, TS, CPM outcomes).

Required clarifications:

• Will PPT, TS, and CPM be analysed as continuous outcomes in ANCOVA/LMM?

• Will outliers (common in QST data) be handled through transformation or robust methods?

• How will bilateral PPT sites be modelled (averaged, modelled separately, or nested within subject)?

7. Per-Protocol vs. Intention-to-Treat

While ITT analysis is planned, the protocol does not describe conditions for per-protocol analysis or adherence thresholds.

Recommendation:

Specify whether:

• A per-protocol set will also be reported.

• Exercise dose or session attendance will be incorporated into sensitivity analyses.

Minor Comments

1. Effect Size Reporting

Partial eta-squared is appropriate, but confidence intervals for effect sizes should also be reported when possible.

2. Covariate Inclusion

Adjusting only for sex, baseline FIQR, and baseline outcome may be insufficient if strong prognostic variables exist (e.g., medication use, FM symptom severity, kinesiophobia). Consider prespecifying additional adjustment factors or providing a rationale for the limited covariate set.

3. Rescue Medication

The plan instructs participants to avoid analgesics 24 hours before assessments. Clarify whether medication use will be recorded and included as a covariate or descriptive variable.

4. Intervention Fidelity Metrics

Session-level variables (e.g., RPE, repetitions, pain pre/post) are being collected but are not included in the analysis plan. Clarify whether these will inform adherence, dose-response exploration, or moderation analysis.

5. Adverse Event Statistical Reporting

Although AEs will be reported descriptively, specify whether incidence rates between groups will be compared statistically (risk ratios or risk differences).

Reviewer #2: Summary

The work is a study protocol of a randomized controlled clinical trial research project. The clinical trial is targeted at potentially improving the nonpharmacologic treatment given to fibromyalgia, a very common disease with a substantial impact on global health care and population. In this study, the IVR is designed to manipulate the vision, making the patient feel as though they are experiencing reduced body movement compared to the actual motion.

Comments

1. According to your 5th reference (Cabo-Meseguer A, Cerdá-Olmedo G, Trillo-Mata JL. Fibromialgia: prevalencia, perfiles epidemiológicos y costes económicos. Med Clin (Barc). 2017;149:441–448. doi:10.1016/j.medcli.2017.06.008), the annual expenditure of fibromyalgia treatment in Spain is more than 12,993 million euros, but on line 55, you stated 4.2 billion; you need to clarify the discrepancy. Furthermore, your 5th reference is focused on Spain, so if you want to mention figures related to the USA, I recommend you use studies done in the USA.

2. The abbreviations EG and CG are used in lines 122 and 125, but the extended versions are mentioned in176 and 177; correct that.

3. In the “Ethical approval and registration” section, I recommend you include a statement that shows that participants can drop out of the study at any time without any fear of being denied any health service at your hospital or other health care facilities.

4. The authors need to mention the reason for stratifying the patients by strata, considering sex and FM severity in the manuscript.

5. TENS, AE, and RPE are included only as abbreviations; include the extended versions as well.

6. Your study is going to use fully immersive VR technology, but you described it as immersive VR. According to the classification, immersive can be fully or semi-immersive. Therefore, for better clarity, you should consistently use the term fully immersive VR.

Questions

1.Wouldn't using different diagnostic criteria for the diagnosis of fibromyalgia affect your statistical findings?

2.What language are you going to use to assess the FIQR; Central Sensitisation Inventory (CSI); Multidimensional Fatigue Inventory (MFI-20); Pittsburgh Sleep Quality Index (PSQI); EQ-5D-5L? and the other scores?

3. Is the presence of comorbidities like hormonal abnormalities, respiratory illnesses profile of the samples going to be assessed?

4.Did you consider concurrent rheumatological disease, such as rheumatoidarthritis or osteoarthritis, as exclusion criteria?

5. What are the parts of the device to be used in the VR treatment?

6. What types of videos or games are going to be used for VR treatment?

7. What are the operational definitions of the “physiological, psychological, and physical outcomes” of fibromyalgia?

8. Why do you want to include only those patients with a pain intensity score of ≥3?

9. What are the comorbidities and/or symptoms that you will consider as  contraindications for IVR and exercise-based interventions?

10. Why did you select 6 weeks for the duration of therapy? And how did you determine the 2 times per week frequency of sessions? And why did you select 60 minutes for the duration of a session?

11. Why did you choose HTC Vive Pro?

12. Why did you choose 20% arm underestimation during flexion and 10% underestimation during extension specifically?

7. PLOS authors have the option to publish the peer review history of their article (what does this mean? ). If published, this will include your full peer review and any attached files.

**Do you want your identity to be public for this peer review?** For information about this choice, including consent withdrawal, please see our Privacy Policy .

Reviewer #1: **Yes:** Dr Shah-Jalal Sarker

Reviewer #2: No

---

## [Author Response · Author response to Decision Letter 1]

25 Mar 2026

RESPONSES TO JOURNAL REQUIREMENTS

JOURNAL REQUIREMENTS

FIRST REQUIREMENT

“Please ensure that your manuscript meets PLOS ONE's style requirements, including those for file naming. The PLOS ONE style templates can be found at

https://journals.plos.org/plosone/s/file?id=wjVg/PLOSOne_formatting_sample_main_body.pdf and https://journals.plos.org/plosone/s/file?id=ba62/PLOSOne_formatting_sample_title_authors_affiliations.pdf”

Response

We have carefully revised the manuscript to ensure compliance with PLOS ONE’s style requirements, using the journal’s formatting sample files for the main body and for the title, authors, and affiliations page as reference guides. Specifically, we reviewed the manuscript structure, title page, author and affiliation formatting, section layout, and file naming, and we amended these elements where needed so that the revised submission conforms to PLOS ONE formatting requirements.

SECOND REQUIREMENT

“Your ethics statement should only appear in the Methods section of your manuscript. If your ethics statement is written in any section besides the Methods, please delete it from any other section.”

Response

We confirm that the ethics statement is presented in the Methods section of the manuscript, in accordance with PLOS ONE’s requirements. We also ensured that the ethics information is reported consistently across the revised submission materials.

THIRD REQUIREMENT

“Please provide a complete Data Availability Statement in the submission form, ensuring you include all necessary access information or a reason for why you are unable to make your data freely accessible. If your research concerns only data provided within your submission, please write "All data are in the manuscript and/or supporting information files" as your Data Availability Statement.”

Response

We have provided a complete Data Availability Statement in the submission form. As this manuscript is a study protocol and does not report study results or generated datasets, the statement clarifies that no datasets were generated or analysed for the current article. It also specifies that, upon trial completion, the de-identified dataset, accompanying data dictionary, and analysis code will be deposited in Zenodo and made publicly available under an open licence, with DOIs to be provided upon release.

FOURTH REQUIREMENT

“When completing the data availability statement of the submission form, you indicated that you will make your data available on acceptance. We strongly recommend all authors decide on a data sharing plan before acceptance, as the process can be lengthy and hold up publication timelines. Please note that, though access restrictions are acceptable now, your entire data will need to be made freely accessible if your manuscript is accepted for publication. This policy applies to all data except where public deposition would breach compliance with the protocol approved by your research ethics board. If you are unable to adhere to our open data policy, please kindly revise your statement to explain your reasoning and we will seek the editor's input on an exemption. Please be assured that, once you have provided your new statement, the assessment of your exemption will not hold up the peer review process.”

Response

We have revised the Data Availability Statement in the submission form to clarify our data-sharing plan in line with PLOS ONE policy. As this manuscript is a study protocol and does not report study results or generated datasets, the revised statement specifies that no datasets were generated or analysed during the current study. It also states that, upon trial completion, the de-identified dataset, accompanying data dictionary, and analysis code will be deposited in Zenodo and made publicly available under an open licence, with the corresponding DOIs to be provided upon release.

FIFTH REQUIREMENT

“We note that the grant information you provided in the ‘Funding Information’ and ‘Financial Disclosure’ sections do not match.

When you resubmit, please ensure that you provide the correct grant numbers for the awards you received for your study in the ‘Funding Information’ section.”

Response

Thank you for highlighting this discrepancy. We acknowledge that the grant information reported in the “Funding Information” and “Financial Disclosure” sections was inconsistent. We have reviewed the funding details and will revise the “Funding Information” section at resubmission so that the correct grant number(s) are provided and both sections are fully aligned.

SIXTH REQUIREMENT

“Thank you for stating the following financial disclosure:

This work was supported by grants from the University CEU Cardenal Herrera (GIR25/41).

Please include this amended Role of Funder statement in your cover letter; we will change the online submission form on your behalf.”

Response

Thank you for this clarification. We confirm that the funder’s role was limited to financial support only. The funders had no role in study design, data collection and analysis, decision to publish, or preparation of the manuscript. As requested, we have included the amended Role of Funder statement in the cover letter.

SEVENTH AND EIGHTH REQUIREMENT

“Please amend your authorship list in your manuscript file to include author Juan Francisco Lisón Párraga”

“Please amend the manuscript submission data (via Edit Submission) to include author Juan Francisco Lisón”

Response

We have reviewed and updated the authorship information across the submission materials to ensure consistency between the manuscript file and the online submission data. In accordance with the author’s preferred publication name, this author is listed as Juan Francisco Lisón in the manuscript, and the submission metadata have been amended accordingly so that the author details are complete, accurate, and fully aligned across the revised submission.

NINTH REQUIREMENT

“Please amend either the abstract on the online submission form (via Edit Submission) or the abstract in the manuscript so that they are identical.”

Response

We have reviewed the abstract in both the manuscript and the online submission form and ensured that the two versions are identical across the revised submission.

TENTH REQUIREMENT

“We note that there is identifying data in the Supporting Information file <S4 Document. Ethics approval (EN).pdf, S5 Document. Ethic approval (ES).pdf>. Due to the inclusion of these potentially identifying data, we have removed this file from your file inventory. Prior to sharing human research participant data, authors should consult with an ethics committee to ensure data are shared in accordance with participant consent and all applicable local laws.

• Name, initials, physical address

• Ages more specific than whole numbers

• Internet protocol (IP) address

• Specific dates (birth dates, death dates, examination dates, etc.)

• Contact information such as phone number or email address

• Location data

• ID numbers that seem specific (long numbers, include initials, titled “Hospital ID”) rather than random (small numbers in numerical order)

Additional guidance on preparing raw data for publication can be found in our Data Policy (https://journals.plos.org/plosone/s/data-availability#loc-human-research-participant data-and-other-sensitive-data) and in the following article: http://www.bmj.com/content/340/bmj.c181.long.

Please remove or anonymize all personal information (<specific identifying information in file to be removed>), ensure that the data shared are in accordance with participant consent, and re-upload a fully anonymized data set. Please note that spreadsheet columns with personal information must be removed and not hidden as all hidden columns will appear in the published file.”

Response

We have reviewed the supporting ethics approval documents and removed non-essential identifying information prior to re-upload. Specifically, personal names and handwritten signatures have been anonymised in the revised files, while retaining the ethics approval information necessary to document institutional approval. The revised supporting files are therefore fully anonymised for public sharing.

RESPONSE TO REVIEWERS

We would like to express our sincere gratitude to both reviewers for the time, dedication, and thoughtful consideration they have devoted to evaluating our manuscript. We truly appreciate the depth and quality of their comments, which have been highly constructive and insightful.

Their observations and recommendations have allowed us to carefully re-examine several aspects of the protocol, clarify key methodological points, improve the precision of our reporting, and strengthen the scientific rationale underlying our decisions. We believe that addressing these comments has significantly enhanced the clarity, transparency, and overall methodological rigor of the manuscript.

We are grateful for the opportunity to revise our work in light of these valuable suggestions, which have undoubtedly contributed to improving the quality and robustness of the article. For ease of review, we note that all references cited throughout the document are fully listed in the final bibliography section.

REVIEWER 1:

MAYOR COMMENTS:

FIRST MAYOR COMMENT

“Sample Size Justification: Effect Size Assumptions Need Clarification

The sample size uses an assumed SD = 20 and an MCID of ≈14% on the FIQR to detect between-group differences using ANCOVA.

However:

• The protocol does not state the absolute expected mean FIQR score on which the 14% MCID is computed.

• It is unclear whether the authors powered the study for between-group mean difference or group × time interaction.

Recommendation:

Explicitly report the assumed baseline FIQR mean, the corresponding absolute MCID value, the expected post-intervention difference, and whether the model used in the power calculation aligns with the final analysis approach (ANCOVA vs. repeated-measures ANCOVA). Provide the full numeric parameters used in G*Power for reproducibility.”

Response

We thank the reviewer for this important methodological observation. We have revised the manuscript to clarify the assumptions and ensure full reproducibility of the sample size calculation.

Specifically, we now report the assumed baseline FIQR mean used to operationalise the ≈14% MCID, and we translate this to an absolute between-group difference. Assuming a baseline mean FIQR score of 65 points, a 14% difference corresponds to 9.1 FIQR points. With an assumed SD of 20, this corresponds to an effect size of d ≈ 0.46 and Cohen’s f = 0.23.

We also clarify that the trial was powered to detect the baseline-adjusted between-group difference at the post-intervention assessment (i.e., not a group × time interaction). Although the a priori calculation was performed using an ANCOVA framework, the primary analysis will use a prespecified two-timepoint linear mixed-effects model; the post-intervention between-group contrast from this model targets the same baseline-adjusted post-intervention comparison as the ANCOVA used for the sample size calculation.

For reproducibility, the G*Power inputs are now explicitly provided: G*Power 3.1.9.2; test family: F tests; statistical test: ANCOVA: fixed effects, main effects and interactions; number of groups: 2; number of covariates: 1 (baseline FIQR); effect size f = 0.23; α = 0.05 (two-sided); power (1–β) = 0.80; allocation ratio 1:1; total N = 64 (32 per group), inflated to N = 80 to allow for 20% dropout.

Accordingly, the revised manuscript text now reads as follows:

“Sample size was calculated using G*Power 3.1.9.2 based on standard methodological guidelines. The primary outcome is the impact of FM on patients' lives, assessed with FIQR. The minimum clinically important difference between groups was set at ≈14%, based on previous research [37]. Assuming a baseline mean FIQR score of 65 points, this corresponds to an absolute between-group difference of 9.1 FIQR points. With α = 0.05 (two-sided), 80% power, and an assumed SD of 20, the trial was powered to detect this baseline-adjusted post-intervention between-group difference using an ANCOVA framework with baseline FIQR as a covariate. This corresponds to d ≈ 0.46 (9.1/20), equivalent to Cohen’s f = 0.23. G*Power inputs were: test family = F tests; statistical test = ANCOVA: fixed effects, main effects and interactions; number of groups = 2; number of covariates = 1 (baseline FIQR); effect size f = 0.23; α = 0.05; power (1−β) = 0.80. This yielded 32 participants per group (64 total). Allowing for 20% dropout, the target sample was increased to 80 participants. This calculation was designed to detect the baseline-adjusted between-group difference in FIQR at the post-intervention assessment and corresponds to the post-intervention between-group contrast estimated from the prespecified two-time point LMM.”

SECOND MAYOR COMMENT

“Choice of Analysis Method: Repeated-Measures ANCOVA May Not Be Appropriate

The plan states that two-way repeated-measures ANCOVA will be used, adjusting for covariates (sex, FIQR level, baseline scores).

Two key concerns:

• Repeated-measures ANCOVA is not a standard or well-defined model.

• Adjusting for baseline within a repeated-measures framework typically leads to statistical redundancy and may violate model assumptions.”

Recommendation:

Use a linear mixed-effects model (LMM), which is the standard for RCTs with repeated measures, missing data handled under MAR, and covariate adjustment. LMMs also allow random intercepts (and slopes if appropriate), avoid sphericity assumptions, and appropriately model within-subject correlation.

Response

We thank the reviewer for this valuable statistical recommendation. We agree that the term “two-way repeated-measures ANCOVA” is not a standard, well-defined framework for longitudinal RCT data and that adjusting for baseline within a repeated-measures ANCOVA structure may be statistically redundant and conceptually unclear.

In response, we have revised the analysis plan to use linear mixed-effects models (LMMs), which are the standard approach for repeated measures in RCTs. The LMM will include fixed effects for group, time, and the group × time interaction, with a participant-specific random intercept to account for within-subject correlation, and adjustment for prespecified covariates (sex and baseline FM severity level, where applicable).

Given that outcomes are assessed at baseline and post-intervention only, the primary treatment effect is the between-group difference in change from baseline to post-intervention (i.e., the group × time interaction). For this two-timepoint setting, we specify a constrained longitudinal data analysis (cLDA) parameterisation (baseline means constrained equal), in which the treatment effect can be expressed as the post-intervention between-group contrast and aligns with the baseline-adjusted ANCOVA estimand used for the sample size calculation.

Accordingly, we have revised the “Statistical methods” section, and the manuscript now reads as follows:

“Primary and secondary outcomes will be analysed using linear mixed-effects models (LMMs), including fixed effects for group, time, and the group × time interaction, with a random intercept for participants to account for within-subject correlation. Outcomes are assessed at baseline and post-intervention only. Models will be adjusted for prespecified covariates (sex an

---

## [Decision Letter · Decision Letter 1]

15 Apr 2026

Effectiveness of a fully immersive virtual reality-based therapeutic exercise programme with altered visual feedback in patients with fibromyalgia: A study protocol for a randomised controlled trial

PONE-D-25-62476R1

Dear Dr. Cuenca,

We’re pleased to inform you that your manuscript has been judged scientifically suitable for publication and will be formally accepted for publication once it meets all outstanding technical requirements.

Kind regards,

Hui-Juan Cao, Ph.D.

Academic Editor

PLOS One

Additional Editor Comments (optional):

Reviewers' comments:

Reviewer's Responses to Questions

**Comments to the Author**

1. Does the manuscript provide a valid rationale for the proposed study, with clearly identified and justified research questions?

Reviewer #1: Yes

2. Is the protocol technically sound and planned in a manner that will lead to a meaningful outcome and allow testing the stated hypotheses?

Reviewer #1: Yes

3. Is the methodology feasible and described in sufficient detail to allow the work to be replicable?

Reviewer #1: Yes

4. Have the authors described where all data underlying the findings will be made available when the study is complete?

Reviewer #1: Yes

5. Is the manuscript presented in an intelligible fashion and written in standard English?

Reviewer #1: Yes

6. Review Comments to the Author

You may also provide optional suggestions and comments to authors that they might find helpful in planning their study.

Reviewer #1: The authors have made substantial improvements to the manuscript. The protocol is now technically sound and suitable for publication. However, the following minor points should be considered to improve clarity and interpretability:

Clarity of Statistical Framework

The description of the LMM and cLDA approach is technically correct but may be difficult for non-statistical readers. A brief plain-language explanation of the primary treatment effect would improve accessibility.

Missing Data Assumption (MAR)

While the analytical approach is appropriate, the manuscript would benefit from a short statement discussing the plausibility and limitations of the missing-at-random assumption in this clinical context.

Secondary Outcomes

Although appropriately designated as exploratory, the large number of secondary outcomes remains extensive. Highlighting a small subset of key secondary outcomes may aid interpretability.

Readability

Some sections—particularly the statistical methods—could be streamlined to improve readability without reducing methodological rigor.

7. PLOS authors have the option to publish the peer review history of their article (what does this mean? ). If published, this will include your full peer review and any attached files.

**Do you want your identity to be public for this peer review?** For information about this choice, including consent withdrawal, please see our Privacy Policy .

Reviewer #1: **Yes:** Dr Shah-Jalal Sarker

---

## [Editor Report · Acceptance letter]

PONE-D-25-62476R1

PLOS One

Dear Dr. Amer-Cuenca,

I'm pleased to inform you that your manuscript has been deemed suitable for publication in PLOS One. Congratulations! Your manuscript is now being handed over to our production team.

Kind regards,

on behalf of

Dr. Hui-Juan Cao

Academic Editor

PLOS One